# The Effect of Face Masks during COVID-19 Pandemic on Ocular Surface Temperature—A Clinical Thermographic Analysis

**DOI:** 10.3390/diagnostics12061431

**Published:** 2022-06-10

**Authors:** Noa Kapelushnik, Shahar Benyosef, Alon Skaat, Amir Abdelkader, Daphna Landau Prat, Sharon Blum-Meirovitch, Ari Leshno

**Affiliations:** 1The Goldschleger Eye Institute, Sheba Medical Center, Ramat Gan 5262100, Israel; kapelushniknoa@gmail.com (N.K.); shaharbenyo@gmail.com (S.B.); askaat11@gmail.com (A.S.); amir.abe86@gmail.com (A.A.); daphna.landau@gmail.com (D.L.P.); sharon.blum@gmail.com (S.B.-M.); 2Sackler Faculty of Medicine, Tel-Aviv University, Tel Aviv 6997801, Israel; 3The Arrow Project for Junior Investigators, Sheba Medical Center, Ramat Gan 5262100, Israel; 4Talpiot Medical Leadership Program, Sheba Medical Center, Ramat Gan 5262100, Israel

**Keywords:** thermography, COVID-19, face mask, ocular surface

## Abstract

Due to the COVID-19 pandemic, the center for disease control and prevention (CDC) recommends face-mask wearing for all people above the age of two years. The wearing of face masks creates a unique airflow towards the ocular surface which may influence the normal physiological conditions of the ocular and periocular surface. Healthy subjects with no ocular history were enrolled in this cross-sectional study. The Therm-App thermal imaging camera was used for ocular surface temperature (OST) measurements during inspirium and expirium. Five regions of interest (ROIs) were used to measure OST: medial conjunctive, cornea, lateral conjunctive, upper eyelid and entire orbital area. Additional measurements in the same locations were made with the upper margin of the mask taped with micropore surgical tape. Thirty-one patients were included in this study. OST during expirium was significantly higher compared to the temperature during inspirium in all locations measured (*p* < 0.001, paired samples *t*-test). The temperature of the upper eyelid was higher by more than 0.5 °C during expirium. Taping the mask’s upper edges to the skin resulted in non-significant temperature changes in inspirium vs. expirium. In conclusion, wearing a face mask creates air flow towards the periocular and ocular surface, which changes the OST mostly on the eyelids.

## 1. Introduction

Since the beginning of the COVID-19 pandemic the center for disease control and prevention (CDC) has recommended face-mask wearing for all people above the age of two years based on publications regarding the efficacy of facial masks in the reduction of COVID-19 transmission and exposure [1,2]. In many countries, regulations were enacted regarding an obligatory duty to wear face mask. As a result, the use of surgical face masks has increased exponentially with millions of people wearing them for long duration throughout the day.

Most of the commonly used face masks (e.g., disposable surgical or reusable fabric masks) are not air-tight superiorly. This configuration results in an unnatural airflow towards the eyelids and ocular surface during expiration. Exhaled air jets coming out from the superior edges of the face mask towards the eyes and periocular surfaces might change the temperature of these areas, as the exhaled air is usually warmer than the environment [3].

Thermoregulation is one of the fundamental physiological processes, crucial for proper tissue function. Mapstone was the first to use infrared (IR) thermography to measure the ocular surface temperature (OST) back in 1968 [4]. However, only in the last years, the interest in ocular temperature has grown, largely thanks to technological improvements that increased the ability to measure ocular temperature more accurately, with higher resolution and speed [5,6,7,8]. For example, older IR thermographers had frame speeds of 4 Hz while many today have speeds up to 60 Hz [5]. In addition, the devices are more affordable and accessible than in the past, allowing more ophthalmology researchers the opportunity to investigate the relation between ocular surface temperature and different pathologies.

Previous studies have shown that changes in ocular temperature might be related to different ophthalmic pathologies such as cataract, presbyopia and discomfort sensation in patients with dry eye disease [9,10,11,12]. Ocular surface temperature (OST) changes may also have an impact on the ocular blood flow and circulation [13]. In a recent large prospective cross-sectional study, Cohen et al., used the Therm-App thermal imaging camera (Opgal Ltd., Karmiel, Israel) to measure OST among individuals undergoing annual general screening tests. In this cohort, they observed a significantly higher OST among persons diagnosed with ischemic heart disease, although no significant association was noted for persons with hypertension, diabetes mellitus or active smoking [14]. The use of OST as a possible biomarker for retinal disease and glaucoma has also been reported recently [15,16]. In addition, there is ample evidence in the literature that SARS-CoV-2 may be transmitted through the eye by either directly infecting cells on the ocular surface, or that the virus resides on the tear film and can be carried by the tears through the nasolacrimal duct and infect the nasal or gastrointestinal epithelium. Therefore, maintaining and preserving the physiologic properties of the ocular surface is also highly important with respect to reducing the risk of SARS-CoV-2 transmission [17,18,19,20,21,22].

Although OST can be an indicator for certain ocular conditions, such as inflammation or ischemic processes, it is also important to remember that it might also be influenced by various extrinsic factors (e.g., ambient temperature, humidity) [23]. Furthermore, these extrinsic factors might also affect tear-film evaporation rate which in itself is an important determinant of OST [24]. Alternatively, abnormal OST might suggest the presence of tear-film pathology [25].

In this study we evaluated the effect of face-mask wear on ocular and periocular temperature. Wearing a face mask that is not air-tight effectively causes constant changes in the periorbital environment through the breathing cycle. We hypothesized that these changes would result in temperature changes of the ocular and periocular region.

## 2. Materials and Methods

### 2.1. Subjects

Healthy volunteers with normal eye examination and a negative history of ocular disease were recruited for the study. All subjects self-reported no history of ocular disease and a systematic review of ocular history was performed for confirmation. In addition, prior to recruitment, all subjects underwent a basic eye exam by slit-lamp biomicroscopy to rule out the presence of any undiagnosed conditions that might have an effect on OST.

Subjects using any form of eye drops (including artificial eye drops), contact lens wear or had any personal history of ocular disease or ocular surgery (other than cataract removal or laser refractive surgery performed at least two years prior to recruitment) were excluded. Subjects suffering from fever, recent illness, or major cardiovascular pathology were excluded as well.

The study followed the tenets of the Declaration of Helsinki and was approved by the Institutional Review Board at Sheba Medical Center. Informed consent was obtained from each study participant after an explanation of the purpose and description of the procedures of the study.

### 2.2. Thermographic Image Capture

Based on a previously reported method for obtaining thermographic ocular images and measurements [26], a standard protocol was maintained in all cases in order to reduce the risk of bias. Thermographic imaging was performed under controlled environmental conditions, with a fixed room temperature of 25 °C, and no air drafts. Thermographic images were taken only after at least 20 min of acclimatization to room temperature.

Prior to image capture, the room and core body temperature (per os) were recorded. Thermal images were taken from the same distance of 30 cm. Video and still thermal images were taken using Therm-App Pro TH (Opgal Optronic Industries Ltd. Karmiel, Israel) camera with a 9 mm lens (384 × 288-pixel resolution).

The subjects were instructed to place their head in the slit lamp, and look straight into the camera. During imaging, subjects were requested to breath normally with their mouth and nose covered by a standard surgical face mask (Sion Biotext medical Ltd., Maastricht, The Netherlands). The images were taken consecutively over a 30 s period, capturing the thermographic data at inspirium and expirium. After completion, 20 of the subjects were also asked to repeat the imaging after sealing the superior end of the mask using micropore surgical tape (3M, Saint Paul, MN, USA). For each participant, the right eye was used for analysis.

### 2.3. OST Measurements

The OST value was retrieved from the thermographic images by means of IRT Cronista 4.0 software (GRAYESS Inc., Bradenton, FL, USA). For each patient, the best thermal images at inspirium and expirium, with and without taping, were selected and analyzed (a total of 4 images per subject, Figure 1).

For each image, 5 regions of interest (ROIs) were used to measure the OST: the medial conjunctive, the cornea, the lateral conjunctive, the upper eyelid and the entire orbital area (Figure 2). The ROIs were marked by a masked observer (SB) and confirmed independently by another masked observer (AL) in order to avoid a measurement bias.

### 2.4. Statistical Analysis

The statistical software SPSS version 25.0 (SPSS, Inc., Chicago, IL, USA) was used for data analysis. Student’s paired *t*-test was used for compression expirium and inspirium OST. Possible confounders were detected using Pearson’s correlation coefficient for continuous variables and univariate logistic regression tests were applied for categorical variables. Statistical significance was set at *p* < 0.05. Minimum sample size for 80% power and 0.05 alpha, was estimated as 27 eyes/subjects using a single-sided sample size calculator for comparing two paired means [27] based on an expected mean of the paired differences of 0.5 and expected 1.0 standard deviation of the paired differences based on previous studies [14,15,16]. Assuming a 10–15% dropout rate, the total number of participants needed was estimated to be 31 subjects.

## 3. Results

A total of 31 eyes of healthy individuals were recruited at the Goldschleger Eye Institute Sheba Medical center, between December 2020 and February 2021. The mean age was 43.6 (Table 1).

### 3.1. Expirium vs. Inspirium

Without taping, the OST was significantly higher during expirium compared to the OST during inspirium, in all of the measured locations (*p* < 0.001). The mean increase in OST from inspirium to expirium at each measured region of interest is depicted in Figure 3. The upper eyelid margin had the greatest temperature change.

Neither sex, age, room temperature or body temperature were found to have a significant effect on measured OST in all ROIs, both while in expirium and inspirium, nor did we observe a significant correlation with the change in OST between the two stages.

### 3.2. The Effect of Taping

During the taping session, the OST measurements were significantly lower in all regions of interest as compared to the measured OST during expirium without taping. This difference was maintained even when the taped thermographic image with the highest OST measurements was used for comparison (Table 2). Conversely, no significant differences were observed between inspiration without taping and inspiration with taping, even when the taped thermographic image with the lowest OST was used for comparison.

## 4. Discussion

This cross-sectional study examined the acute influence of face-mask wearing on ocular surface temperature. As we hypothesized, the use of a standard face mask resulted in significant changes in OST during the breathing cycle, mostly due to the effects of air-jets towards the orbit during expirium. The change was most notable in the eyelid margins, with a mean rise of 0.5 °C during expirium. These changes were eliminated by taping the upper margin of the mask, preventing the airflow to the ocular region.

### 4.1. Extrinsic Determinants of OST

While OST can be a biomarker for certain ocular pathologies such as glaucoma or retinal pathologies [4,15,16], it is important to remember that it is determined by multiple factors, both intrinsic (e.g., body temperature, eyelid closure time, ocular inflammation, tear-film stability, blink frequency, tear volume, etc.) and extrinsic (e.g., ambient environmental temperature and room humidity) [23,25,28].

Although data on the environmental effect on OST is relatively limited, several studies have shown a correlation between both ambient temperature and room humidity to OST [23,24,28,29,30]. However, over all, it seems that ambient temperature has a greater impact on OST than room humidity, at least for single-point OST measurements [31]. More importantly, Abusharha et al., showed that ambient temperature has a considerable effect on human tear-film characteristics. Using a controlled environment chamber, they were able to show that an increase in ambient temperature not only results in a significant change in OST but also in increased tear evaporation rates and lipid layer thickness [30]. Herein, we attempted to evaluate the effect of the environmental changes induced by respiration while wearing a standard face-mask. Contrary to Abusharha et al., our goal was to simulate real-life situations therefore we did not attempt to control the exact extent of these changes. However, we also found that even transient environmental changes can induce a change in single-point OST measurements.

### 4.2. Face-Mask-Induced OST Changes Might Affect the Tear Film

The changes in OST observed herein, secondary to the use of face masks without taping, might also have a secondary effect on the ocular surface by inducing changes in the tear film. Several studies investigated the association between OST and of ocular surface abnormalities, mostly with regards to dry eye disease (DED) and tear-film abnormalities. The properties of the tear film consist of multiple factors including production rate, composition, distribution and tear turnover, all of which are crucial to maintain tear homeostasis. One of the most important biomarkers for DED is tear-film instability, which can be measured by tear-film breakup time (TBUT). Not surprisingly, TBUT in itself is one of the most important determinants of OST [31]. In a study by Purslow et al., dynamic ocular thermography was used to measure the continuous OST for 8 s after blinking of 25 eyes. Tear break up time was assessed using a tearscope. They found that individuals with poor tear-film stability had a higher ocular surface temperature (higher OST) compared to those with intact tear-film stability [25]. Kamao et al., used an ocular surface thermographer to measure the OST of 30 patients with the diagnosis of dry eye compared with 30 healthy patients. The measured temperature correlation with tear-film break up time, Schirmer test and fluorescein staining scores was determined. They reported that even a slight gradient in temperature correlates with an increase in tear break up time [32]. Similarly, Morgan et al., detected higher OST at the center of the cornea among individuals suffering from dry eye disease (DED). In their study, a thermogram was used to assess the ocular surface temperature of 36 patients diagnosed with dry eye disease [33]. Craig et al., found that the temperature differential was higher in dry eyes [34]. Although in both cases, the changes in OST were attributed to ocular pathology (e.g., fast tear-film evaporation, ocular surface inflammation, etc.), it is possible that the extrinsic elevation of OST, such as a warm air jet towards the eyes during expirium, might have the same effect. However, it is important to note that reports on the association between OST and DED vary considerably with contradictory results throughout the literature, as described by Shah and Galor [31].

It is worth noting that ocular warming, mostly of the eyelids, has been recommended to treat DED secondary to meibomian gland dysfunction (MGD). In such cases, the delivery of the lipid layer in the tear film is impaired, resulting in reduced tear-film stability and increased evaporation, which leads to the development of signs and symptoms of dry eye [35,36,37,38,39,40,41]. The idea is to facilitate the melting of the thickened secretions subsequently restoring the natural meibum in the tear-film lipid layer [42,43]. Wang et al., recently described the positive influence of warming the orbit and ocular surface on tear-film stability and lipid layer grade [44]. In their study, a heating device was used to raise OST and compare the changes in TBUT before and after heating. Although heating would be expected to increase tear film evaporation, as OST increased, the TBUT was prolonged, presumably thanks the increased level of lipids in the tear film. Based on this observation, it might be possible that the increased OST secondary to the face mask might also have a positive effect in patients with MGD. However, the duration of exposure to the high and steady temperature might also be considered. Contrary to the constant rise in temperature generated by a warming device or even warm compresses, the rise in OST from the use of a face mask is transient and dependent on the breathing cycle (warmer during expiration and colder during inhalation). Thus, it might not be enough to restore the meibum in the tear film. If so, the heating might only result in faster evaporation of the tear film, and therefore worsening of the DED symptoms.

In any case, the connection between tear-film properties and OST seems to be bi-directional and the two show a symbiotic relationship. Furthermore, while certain changes in OST might prove beneficial for patients with certain conditions, it might prove to produce the opposite effect with other DED-related etiologies and the management should be tailored appropriately.

### 4.3. Use of Face-Mask and Ocular Pathology

It is important to note that while changes in ocular surface temperature might be significant, the change in ocular temperature alone is not solely responsible for the rise in complaints of dry eye during the COVID-19 pandemic, and that the reason is most likely multifactorial. Previous studies show that during the COVID-19 pandemic, ocular complaints such as red eye, irritation, tearing and eye discomfort have become common. Several studies suggested that the rise in ocular complaints is mostly part of dry eye disease, and found that these complaints correlate with a long duration of mask use, hence defining a new condition—“mask-associated dry eye” (MADE) [43,44,45,46,47,48]. Krolo et al.’s study is a prospective cohort study included 203 participants. An ocular surface disease index score was evaluated using a questionnaire, as well as sex, age, duration of face-mask wear and prior dry eye disease. Their results indicated that participants wearing a face mask for more than 3 h a day demonstrated a significantly higher ocular surface disease index score and thus, confirming the existence of MADE [48].

MADE has been attributed to increased virus transmission both from frequent eye touching and air flow. In Hadayer et al.’s study, the periocular area was inspected for air leak during normal respiration, speech, and deep respiration using thermal cameras. [49] They found that during surgical mask wear, air jets radiate towards the eyes, allowing dispersion of bacteria from the oral cavity to the ocular region. This is also supported by the work of Raevis et al.; [50] in their study, the air flow was evaluated using schlieren imaging, which showed air jets escaping the upper aspect of the face mask. Tape covering the superior aspects of the mask resulted in air flow towards the eye.

Other clues for the mask as a mitigator of breath-to-eye infections include SARS-CoV-2-associated conjunctivitis that is speculated to come from the breath via masks to the eye [51,52,53]. Additionally, a documented rise in the incidence of chalazion correlated to face-mask wearing compliance was reported by Silkiss et al [54]. In their retrospective multicenter study, all of the patients diagnosed with chalazion or hordeolum between January and August 2020 were identified. The incidence of these diagnoses in the time period was compared to the incidence in the prior year. Their results indicated that the incidence of chalazion diagnosis rose significantly during COVID-19 pandemic, most likely due to face-mask wearing.

The observed changes in OST in our study might also contribute to the pathophysiology of MADE and other mask-related ocular conditions. Similar to the previously mentioned studies, we assumed that the mechanism by which face masks affect the OST is probably much contributed to by the warm air jets radiating towards the ocular environment, significantly elevating the OST, most notably on the upper eyelids. This hypothesis is supported by our findings which show no temperature rise when the upper edge of the face mask is covered with tape, preventing the air jet to reach the ocular region. Although a change of only 0.5 °C might be considered small, other studies, such as that of Fabiani et al., demonstrated that a similar amount of change in temperature influenced the intra ocular pressure [55]. This change might also facilitate the spread or infection of certain viruses or bacteria transmitted through the very same air jets. It might also influence other ocular surface factors such as tear break up time and corneal sensation, for example.

Therefore, we believe that patients diagnosed with MADE, or those who are experiencing a worsening of DED-related symptoms should be recommended to perform a trial using a mask with better superior fitness, or even taping of the superior area of the mask, to see if reducing the airflow to the orbit during expirium might obviate the symptoms. As mentioned previously, a subset of patients suffering from MGD might even have experienced an improvement of symptoms with the use of a face mask, secondary to the warming of the eyelids and improved gland function. However, due to the possible transmission of SARS-CoV-2 through the ocular surface, even in such cases it is important to ensure that the use of the face-mask and air-jets to the orbit do not compromise the stability of the tear film and protective layers of the eye from the virus.

### 4.4. Limitations

This study does have several limitations. First, we only evaluated short-term changes in OST. Although it seems likely that the effect of expirium on OST remains similar, longer prospective studies might detect some form of balancing physiologic changes that counteract this effect. Our study included only healthy subjects without any ocular surface disease, therefore we cannot determine whether the effect would be more or less significant among patients suffering from dry eye disease, tear-film abnormalities or similar conditions. In addition, we did not evaluate changes in TBUT that in itself might be related to the use of face masking, as well as having an effect on OST. It should also be mentioned that the measured temperatures herein might be different from previous reports, due to technical differences in thermal imaging devices, as described previously in a review by Shah and Galor [24]. In order to avoid measuring bias, we focused on the change in temperature rather than the absolute values. Although we evaluated the most commonly used type of face mask, there is a wide variety of commercially available masks in which face alignment, especially around the nose, may be different, resulting in greater or lesser changes in OST. The greatest temperature change was demonstrated at the eyelid margin, yet it is important to note that the eyelashes usually outstand as colder than the rest of the orbit. Hong tan et al. [56] described a method to isolate the effect on the skin by eliminating the hairs, however, unfortunately we were unable to implement it herein.

## 5. Conclusions

In summary, our results suggest that wearing commonly used surgical face masks can cause changes in OST and changes of the periocular temperature. These changes seem to result mainly from jets of warm air during expirium. This may be one of the underlying explanations for the development of dry eye disease associated with face-mask wear, as well as other ocular complaints possibly related to the use of face masks.

## Figures and Tables

**Figure 1 diagnostics-12-01431-f001:**
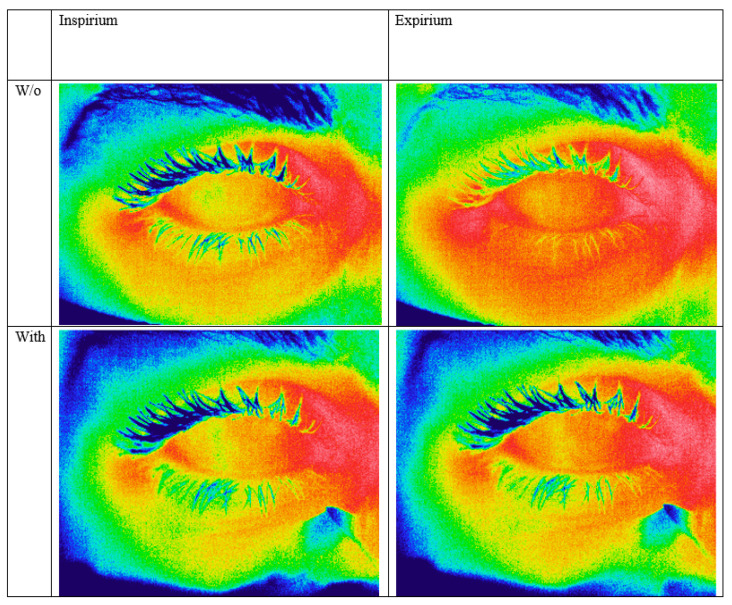
Representative case showing the change in OST between inspirium (**left column**) and expirium (**right column**) with regular wear of facial mask (**first row**) and with superior tapping of the facial mask (**second row**).

**Figure 2 diagnostics-12-01431-f002:**
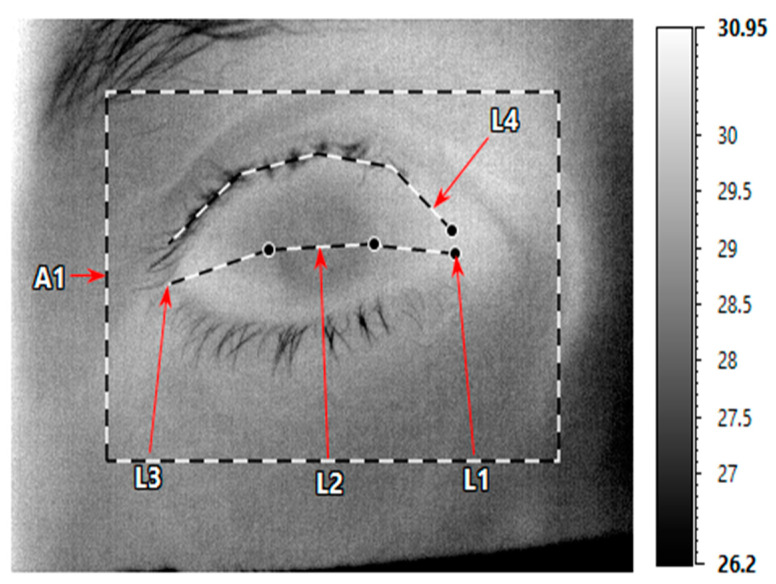
Postacquisition ocular surface temperature (OST) manual measurements performed using of IRT Cronista 4.0 software (GRAYESS Inc., Bradenton, FL, USA). The mean OST was measured along three lines representing the medial (L1), corneal (L2) and lateral (L3) regions of the ocular surface. A fourth line was used to measure the temperature across the eyelid margins (L4). The temperature of the entire orbital surface was also measured (A1).

**Figure 3 diagnostics-12-01431-f003:**
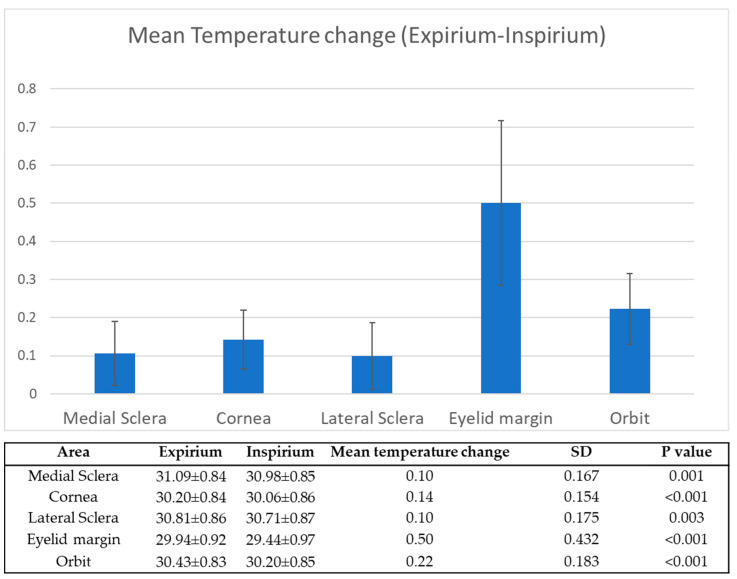
Mean change in OST (Celsius) between in expiration and inspiration while wearing a facial mask.

**Table 1 diagnostics-12-01431-t001:** Study patients’ demographics and environmental measurements.

Number of Patients (Eyes)	31 (31)
Age (years), mean SD	43.6 ± 14.8
Female, n (%)	21 (67.7%)
Body temp, mean SD	36.8 ± 0.34
Room temp, mean (SD)	22.1 ± 1.2

**Table 2 diagnostics-12-01431-t002:** Expirium ocular surface temperature compared to highest temperature measurements with upper mask edge taped.

Area	Expirium No Tape	Taped Highest Measurements	Mean Temperature Change	SD	*p* Value
**Medial sclera**	31.13 ± 0.83	30.99 ± 0.84	0.14	0.26	0.034
**Cornea**	30.28 ± 0.85	30.15 ± 0.79	0.13	0.26	0.037
**Lateral sclera**	30.86 ± 0.88	30.59 ± 1.03	0.27	0.43	0.013
**Eyelid margin**	29.97 ± 0.96	29.56 ± 1.06	0.41	0.60	0.006
**Orbit**	30.47 ± 0.85	30.26 ± 0.83	0.20	0.23	0.001

## Data Availability

Data is available upon request from authors.

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
