# Peer review of "The Effect of Face Masks during COVID-19 Pandemic on Ocular Surface Temperature—A Clinical Thermographic Analysis"

_diagnostics, 2022, doi:10.3390/diagnostics12061431_

Round 1

Reviewer 1 Report

1. Show the characteristics of the participants as a Table.

2. Authors should show how to determine the sample size in the Method.

3. Show the real temperature for two stages or taping or not but not only change in figure 3 and Table 1. 

4. Authors did not examine TBUT in this study. Therefore, remove 4.2 in the discussion section.

Author Response

Reviewer #1 comments

  1. Show the characteristics of the participants as a Table.

The following table has been added to the results:

Table 1. Study patients Demographics and environmental measurements

Number of patients/ eyes

31

Age (years), mean SD

43.6±14.8

Female, n(%)

21 (67.7%)

Body temp, mean SD

36.8±0.34

Room temp, mean (SD)

22.1±1.2

  1. Authors should show how to determine the sample size in the Method.

The following has been added to the methods section:

“Minimum sample-size for 80% power and 0.05 alpha, was estimated as 27 eyes/ subjects using a single-sided Sample Size Calculator for Comparing Two Paired Means (Statulator, 2014, http://statulator.com/SampleSize/ss2PM.html) based on an expected mean of the paired differences of 0.5 and expected 1.0 standard deviation of the paired differences. We recruited 4 additional subjects in case some of the images would be found unusable.”

  1. Show the real temperature for two stages or taping or not but not only change in figure 3 and Table 1.

The data has been added to the tables as requested

  1. Authors did not examine TBUT in this study. Therefore, remove 4.2 in the discussion section.

As the reviewer comments, we did not perform TBUT in this study. However, as the evidence in the literature shows, there is a significant and close association between ocular surface temperature and tear-film abnormalities. In section 4.2 provide a short but detailed review of the most current literature on this association, in an attempt to show how the observed OST changes in our study, as a result from using a face-mask, might at least in part explain, why there has been a rise on symptoms and worsening of ocular surface disease seen since the use of face mask was increase in the covid era.

We agree that the headline for this section might be misleading to some readers. Therefore we have decided to revise the section to better clarify our intent

4.2 ”Face-mask induced OST changes might affect the tear-film

“The changes in OST observed herein, secondary to the use of face masks without tapping might also have a secondary effect on the ocular surface by inducing changes in the tear-film. Several studies investigated the association between OST and of ocular surface abnormalities, mostly with regards to dry eye disease (DED)

two show a symbiotic relationship. Furthermore, while certain changes in OST might prove beneficial for patients with certain conditions, it might prove to produce the opposite effect with other DED related etiologies and the management should be tailored appropriately.”

We also addressed this issue in the limitations section.

“…In addition, we did not evaluated changes in TBUT that in itself might be related to the use of face masking, as well as have an effect on OST.”

Reviewer 2 Report

I read with great interest the work entitled  ‘The effect of face masks during COVID-19 pandemic on ocular surface temperature- A clinical thermographic analysis’. The  study evaluated the effect of face mask wear on ocular and periocular temperature

I ask the authors some small but fundamental clarifications to make the paper publishable.

Major Concerns

  1. In the introduction it should be noted that with regard to SARS-Cov-2 at least two possible routes of entry through the eye have been hypothesized. Hence the need to preserve the physiological protections of the organ.
  2. It should also be remembered the mechanism that allows to maintain ocular temperature including tear turnover. [Napoli, Pietro Emanuele, et al. "The bull’s eye pattern of the tear film in humans during visual fixation on en-face optical coherence tomography." Scientific reports 9.1 (2019): 1-9.]
  3. Is the selection of the sample made on an anamnestic basis or did the patients undergo a preliminary eye examination?
  4. Have the repeatability and reproducibility of the method been evaluated? [cfr Napoli, Pietro Emanuele, et al. "Repeatability and reproducibility of post-mortem central corneal thickness measurements using a portable optical coherence tomography system in humans: A prospective multicenter study." Scientific Reports 10.1 (2020): 1-9.]. And the presence of operator dependent bias (measurement and interpretation of data)?
  5. The results described by the authors appear to be statistically significant. What are the repercussions of the work from a practical point of view? Do the authors have to propose any solutions to obviate exposure to the thermal insult?

Minor concerns

  • The bibliography should be expanded.
  • The article must be reformatted according to the indications provided by the journal.

Author Response

Reviewer #2 comments

I read with great interest the work entitled ‘The effect of face masks during COVID-19 pandemic on ocular surface temperature- A clinical thermographic analysis’. The study evaluated the effect of face mask wear on ocular and periocular temperature

I ask the authors some small but fundamental clarifications to make the paper publishable.

Major Concerns

  1. In the introduction it should be noted that with regard to SARS-Cov-2 at least two possible routes of entry through the eye have been hypothesized. Hence the need to preserve the physiological protections of the organ.

We thank the reviewer for this important comment. The following text has been added to the introduction:

“There is ample evidence in the literature that SARS-CoV-2 may be transmitted through the eye by either directly infecting cells on the ocular surface, or that the virus resides on the tear film and can be carried by the tears through the nasolacrimal duct and infect the nasal or gastrointestinal epithelium. Therefore, maintaining and preserving the physiologic properties of the ocular surface is also highly important with respect to reducing the risk of SARS-CoV-2 transmission.”

  1. It should also be remembered the mechanism that allows to maintain ocular temperature including tear turnover. [Napoli, Pietro Emanuele, et al. "The bull’s eye pattern of the tear film in humans during visual fixation on en-face optical coherence tomography." Scientific reports 9.1 (2019): 1-9.]

We thank the reviewer for this comment. The following was added to the text regarding the importance of tear turnover as well as tear clearance rate:

“The properties of the tear film consist of multiple factors including production rate, composition, distribution and tear turnover, all of which are crucial to maintain tear homeostasis.”

  1. Is the selection of the sample made on an anamnestic basis or did the patients undergo a preliminary eye examination?

All subjects personally reported no history of ocular disease and a systematic review of ocular history was performed for confirmation. In addition, prior to recruitment, all subjects underwent a basic eye exam by slit-lamp biomicroscopy to rule out the presence of any undiagnosed conditions that might have an effect on OST.

The following text was added to the method to clarify this issue:

“All subjects self reported no history of ocular disease and a systematic review of ocular history was performed for confirmation. In addition, prior to recruitment, all subjects underwent a basic eye exam by slit-lamp biomicroscopy to rule out the presence of any undiagnosed conditions that might have an effect on OST.”

  1. Have the repeatability and reproducibility of the method been evaluated? [cfr Napoli, Pietro Emanuele, et al. "Repeatability and reproducibility of post-mortem central corneal thickness measurements using a portable optical coherence tomography system in humans: A prospective multicenter study." Scientific Reports 10.1 (2020): 1-9.]. And the presence of operator dependent bias (measurement and interpretation of data)?

The thermographic image acquisition methods used in our study were based on a standardized protocol that has been used in various previous studies by us and other. The measurements were performed by a masked observer (SBY) and confirmed by an additional masked observer (AL) to avoid operator bias.

The reproducibility of this protocol was validated by Konieczka et al. We added the following to the methods section to clarify this issue:

Based on a previously reported method for obtaining thermographic ocular images and measurements2 a standard protocol was maintained in all cases in order to reduce the risk of bias…”

And later in the text:

“…For each image, 5 regions of interest (ROIs) were used to measure the OST: the medial conjunctive, the cornea, the lateral conjunctive, the upper eyelid and the entire orbital area (Figure 2). The ROIs were marked by a masked observer (SB) and confirmed independently by another masked observer (AL) in order to avoid a measurement bias.”

  1. The results described by the authors appear to be statistically significant. What are the repercussions of the work from a practical point of view? Do the authors have to propose any solutions to obviate exposure to the thermal insult?

We believe that since the change in OST might have both negative and positive influence on ocular symptoms, the decision on management should be individually tailored to match the patients needs and condition. In particular, patients who report worsening of symptoms, we suggest performing a trial using mask with better superior fitness, or even taping of the superior mask, to see if reducing the airflow to the orbit during expirium might obviate the symptoms. Conversely, we also encourage to further explore possible benefits from using the face mask in patients with dry eye disease secondary to meibomian gland disfunction and associated abnormalities in the lipid layer. As mentioned in the discussion, these patients are known to benefit from the use of warm compresses for the eyelids so presumably a similar effect might be obtained from the air-jets toward the orbit with the use of face masks. However, due to the possible transmission of SARS-CoV-2 through the ocular surface, strict percussion must be taken to ensure that the use of the face-mask and air-jets to the orbit do not compromise the stability of the tear film and protective layers of the eye from the virus.

The following was added to section 4.3 of the manuscript:

 “…Therefore, we believe that patients diagnosed with MADE, or those who are experiencing worsening of DED related symptoms should be recommended to perform a trial using mask with better superior fitness, or even taping of the superior mask, to see if reducing the airflow to the orbit during expirium might obviate the symptoms. As mentioned previously, a subset of patients suffering from MGD might have even experience an improvement of symptoms with the use of a face mask, secondary to the warming of the eyelids and improved gland function. However, , due to the possible transmission of SARS-CoV-2 through the ocular surface, even in such cases it is important to ensure that the use of the face-mask and air-jets to the orbit do not compromise the stability of the tear film and protective layers of the eye from the virus.”

Minor concerns

  1. The bibliography should be expanded.

We performed a thorough review of the literature and added any relevant articles that we have found.

  1. The article mu

    Reviewer #2 comments

    I read with great interest the work entitled ‘The effect of face masks during COVID-19 pandemic on ocular surface temperature- A clinical thermographic analysis’. The study evaluated the effect of face mask wear on ocular and periocular temperature

    I ask the authors some small but fundamental clarifications to make the paper publishable.

    Major Concerns

    1. In the introduction it should be noted that with regard to SARS-Cov-2 at least two possible routes of entry through the eye have been hypothesized. Hence the need to preserve the physiological protections of the organ.

    We thank the reviewer for this important comment. The following text has been added to the introduction:

    “There is ample evidence in the literature that SARS-CoV-2 may be transmitted through the eye by either directly infecting cells on the ocular surface, or that the virus resides on the tear film and can be carried by the tears through the nasolacrimal duct and infect the nasal or gastrointestinal epithelium. Therefore, maintaining and preserving the physiologic properties of the ocular surface is also highly important with respect to reducing the risk of SARS-CoV-2 transmission.”

    1. It should also be remembered the mechanism that allows to maintain ocular temperature including tear turnover. [Napoli, Pietro Emanuele, et al. "The bull’s eye pattern of the tear film in humans during visual fixation on en-face optical coherence tomography." Scientific reports 9.1 (2019): 1-9.]

    We thank the reviewer for this comment. The following was added to the text regarding the importance of tear turnover as well as tear clearance rate:

    “The properties of the tear film consist of multiple factors including production rate, composition, distribution and tear turnover, all of which are crucial to maintain tear homeostasis.”

    1. Is the selection of the sample made on an anamnestic basis or did the patients undergo a preliminary eye examination?

    All subjects personally reported no history of ocular disease and a systematic review of ocular history was performed for confirmation. In addition, prior to recruitment, all subjects underwent a basic eye exam by slit-lamp biomicroscopy to rule out the presence of any undiagnosed conditions that might have an effect on OST.

    The following text was added to the method to clarify this issue:

    “All subjects self reported no history of ocular disease and a systematic review of ocular history was performed for confirmation. In addition, prior to recruitment, all subjects underwent a basic eye exam by slit-lamp biomicroscopy to rule out the presence of any undiagnosed conditions that might have an effect on OST.”

    1. Have the repeatability and reproducibility of the method been evaluated? [cfr Napoli, Pietro Emanuele, et al. "Repeatability and reproducibility of post-mortem central corneal thickness measurements using a portable optical coherence tomography system in humans: A prospective multicenter study." Scientific Reports 10.1 (2020): 1-9.]. And the presence of operator dependent bias (measurement and interpretation of data)?

    The thermographic image acquisition methods used in our study were based on a standardized protocol that has been used in various previous studies by us and other. The measurements were performed by a masked observer (SBY) and confirmed by an additional masked observer (AL) to avoid operator bias.

    The reproducibility of this protocol was validated by Konieczka et al. We added the following to the methods section to clarify this issue:

    Based on a previously reported method for obtaining thermographic ocular images and measurements2 a standard protocol was maintained in all cases in order to reduce the risk of bias…”

    And later in the text:

    “…For each image, 5 regions of interest (ROIs) were used to measure the OST: the medial conjunctive, the cornea, the lateral conjunctive, the upper eyelid and the entire orbital area (Figure 2). The ROIs were marked by a masked observer (SB) and confirmed independently by another masked observer (AL) in order to avoid a measurement bias.”

    1. The results described by the authors appear to be statistically significant. What are the repercussions of the work from a practical point of view? Do the authors have to propose any solutions to obviate exposure to the thermal insult?

    We believe that since the change in OST might have both negative and positive influence on ocular symptoms, the decision on management should be individually tailored to match the patients needs and condition. In particular, patients who report worsening of symptoms, we suggest performing a trial using mask with better superior fitness, or even taping of the superior mask, to see if reducing the airflow to the orbit during expirium might obviate the symptoms. Conversely, we also encourage to further explore possible benefits from using the face mask in patients with dry eye disease secondary to meibomian gland disfunction and associated abnormalities in the lipid layer. As mentioned in the discussion, these patients are known to benefit from the use of warm compresses for the eyelids so presumably a similar effect might be obtained from the air-jets toward the orbit with the use of face masks. However, due to the possible transmission of SARS-CoV-2 through the ocular surface, strict percussion must be taken to ensure that the use of the face-mask and air-jets to the orbit do not compromise the stability of the tear film and protective layers of the eye from the virus.

    The following was added to section 4.3 of the manuscript:

     “…Therefore, we believe that patients diagnosed with MADE, or those who are experiencing worsening of DED related symptoms should be recommended to perform a trial using mask with better superior fitness, or even taping of the superior mask, to see if reducing the airflow to the orbit during expirium might obviate the symptoms. As mentioned previously, a subset of patients suffering from MGD might have even experience an improvement of symptoms with the use of a face mask, secondary to the warming of the eyelids and improved gland function. However, , due to the possible transmission of SARS-CoV-2 through the ocular surface, even in such cases it is important to ensure that the use of the face-mask and air-jets to the orbit do not compromise the stability of the tear film and protective layers of the eye from the virus.”

    Minor concerns

    1. The bibliography should be expanded.

    We performed a thorough review of the literature and added any relevant articles that we have found.

    1. The article must be reformatted according to the indications provided by the journal.

    With the help of the journal’s assistant editor the article was reformatted as indicated.

    st be reformatted according to the indications provided by the journal.

With the help of the journal’s assistant editor the article was reformatted as indicated.

Round 2

Reviewer 1 Report

For the sample size calculation, authors should add the reference for determining the difference and SD.

In addition, authors may set the drop rate for the sample size calculation, otherwise you cannot add additional 4 participants in this study.

Author Response

Data from our previous studies has been used to estimate the the difference and SD for the sample size calculation. The following references were added in the appropriate location:

  1. Cohen GY, Ben-David G, Singer R, et al. Ocular surface temperature: Characterization in a large cohort of healthy human eyes and correlations to systemic cardiovascular risk factors. Diagnostics. 2021;11(10). doi:10.3390/DIAGNOSTICS11101877
  2. Leshno A, Stern O, Barkana Y, et al. Ocular surface temperature differences in glaucoma. European journal of ophthalmology. Published online 2021. doi:10.1177/11206721211023723
  3. Naidorf-Rosenblatt H, Landau-Part D, Moisseiev J, et al. OCULAR SURFACE TEMPERATURE DIFFERENCES IN RETINAL VASCULAR DISEASES. Retina (Philadelphia, Pa). 2022;42(1):152-158. doi:10.1097/IAE.0000000000003278

Based on a drop rate of 10-15% the total number of participants was estimated as 31 eyes.